# Data integration uncovers the metabolic bases of phenotypic variation in yeast

**Marianyela Sabina Petrizzelli**[1,2,3,4]*, **Dominique de Vienne**[1], **Thibault Nidelet**[5], **Camille Noûs**[6], **Christine Dillmann**[1]

1 Université Paris-Saclay, INRAE, CNRS, AgroParisTech, GQE–Le Moulon, Gif-sur-Yvette, France, 2 Institut Curie, PSL Research University, Paris, France, 3 INSERM, U900, Paris, France, 4 CBIO-Centre for Computational Biology, MINES ParisTech, PSL Research University, Paris, France, 5 SPO, INRAE, Montpellier SupAgro, Université de Montpellier, Montpellier, France, 6 Laboratoire Cogitamus, France

* marianyela.petrizzelli@curie.fr

**Data Availability Statement:** The datasets and computer codes produced in this study are available at \textit{figshare}, DOI:10.6084/m9. figshare.10266332.

## Abstract

The relationship between different levels of integration is a key feature for understanding the genotype-phenotype map. Here, we describe a novel method of integrated data analysis that incorporates protein abundance data into constraint-based modeling to elucidate the biological mechanisms underlying phenotypic variation. Specifically, we studied yeast genetic diversity at three levels of phenotypic complexity in a population of yeast obtained by pairwise crosses of eleven strains belonging to two species, *Saccharomyces cerevisiae* and *S. uvarum*. The data included protein abundances, integrated traits (life-history/fermentation) and computational estimates of metabolic fluxes. Results highlighted that the negative correlation between production traits such as population carrying capacity ($K$) and traits associated with growth and fermentation rates ($J_{max}$) is explained by a differential usage of energy production pathways: a high $K$ was associated with high TCA fluxes, while a high $J_{max}$ was associated with high glycolytic fluxes. Enrichment analysis of protein sets confirmed our results.

This powerful approach allowed us to identify the molecular and metabolic bases of integrated trait variation, and therefore has a broad applicability domain.

## Author summary

The integration of data at different levels of cellular organization is an important goal in computational biology for understanding the way the genotypic variation translates into phenotypic variation. Novel profiling technologies and accurate high-throughput phenotyping now allows genomic, transcriptomic, metabolic and proteomic characterization of a large number of individuals under various environmental conditions. However, the metabolic fluxes remain difficult to measure. In this work, we take advantage of recent advances in genome-scale functional annotation and constraint-based metabolic modeling to provide a mathematical framework that allows to estimate internal cellular fluxes from protein abundances and elucidate the biological mechanisms underlying phenotypic variation. Applied to yeast as a model system, this approach highlights that the negative

**Funding:** MSP was funded with a public Ph.D. grant from the French National Research Agency (ANR) as part of the Investissement d'Avenir program, through the Initiative Doctorale Interdisciplinaire (IDI) 2015 project funded by the Initiative d'Excellence (IDEX) Paris-Saclay, ANR-11-IDEX-0003-02. The funders had no role in study design, data collection and analysis, decision to publish, or preparation of the manuscript.

**Competing interests:** The authors have declared that no competing interests exist.

correlation between production traits such as maximum population size and growth and fermentation traits is explained by a differential usage of energy production pathways. The ability to identify molecular and metabolic bases of the variation of integrated traits through population studies has a broad applicability domain.

## Introduction

Phenotypic diversity within the living world results from billions of years of evolution. Most evolutionary pressures like mutation, random genetic drift, migration and recombination shape phenotypic diversity by directly changing the genetic composition of populations. The effects of selection are more difficult to predict because fitness is determined by phenotype, which results from a complex interaction between genotype and the environment [1]. An additional layer of complexity results from the fact that life-history traits [2] are the results of processes that occur at the cellular level. During the last decades, there has been a growing interest for a better understanding in evolutionary biology of the so-called genotype-phenotype map (see *e.g.* [3]). In parallel, novel profiling technologies and accurate high-throughput phenotyping strategies have led to the genome-scale characterization of genomic sequences as well as to the quantification of transcriptomic, proteomic and metabolomic data at the individual level. Linking cellular processes to high-level phenotypic traits is becoming a new discipline in Biology, known as integrative biology.

Unicellular organisms are the model species of choice for integrative biology because most of their integrated traits are the direct product of cell metabolism, without needing to take into account the complexity of tissues and organs as in multicellular organisms. Schematically, cells sense the environment and transfer the information via signal transduction chains that interact with gene regulation networks. Gene regulatory networks modulate transcription, translation and post-translational modifications in response to environmental signals, resulting in variations in protein abundances. Differential abundances of enzymatic proteins affect the fluxes of matter and energy that are related to phenotypic traits, including life-history traits and fitness. Thus, in unicellular organisms, five integration levels are usually considered: genomic, transcriptomic, proteomic (including post-translational modifications), metabolic and cellular. The last level is the most integrated, and it encompasses a variety of traits that are more or less related to fitness.

While technical progress has now allowed for genomic, transcriptomic, proteomic and trait levels to be readily measurable in a high number of individuals, metabolic fluxes are still difficult to measure. Although Metabolic Flux Analysis, based on Nuclear Magnetic Resonance (NMR) and differential usage of radioactive isotopes, is powerful [4], it remains low throughput and cannot be applied to a high number of individuals. Technical developments in mass spectrometry have boosted metabolomics [5] by enabling the characterization of the metabolome. However, the technique still suffers from standardization difficulties [6].

Taking advantage of recent advancements in genome-scale functional annotation, constraint-based metabolic models provide a mathematical framework that allows us to predict internal cellular fluxes from *a priori* knowledge of thermodynamic constraints on individual enzymatic reactions, steady state hypotheses and the genome-scale stoichiometry matrix of all metabolic reactions. The idea is to explore system's properties at a steady state, during which internal metabolites stay at a constant concentration while exchange fluxes are constant and correspond to a constant import/export rate. However, because the number of reactions is higher than the number of metabolites, the system has an infinite number of solutions. Flux

Balance Analysis [7, 8] consists in choosing, among all possible solutions, the solution that maximizes the biomass pseudo-flux that represents the conversion rate of biomass precursors into biomass. This method is questionable because evolution is not always based on optimization principles [9]. However, it has been shown to be relevant in some cases, such as chemostat cultures of *Escherichia coli* [10]. Data-driven methods have also been proposed, which consist in choosing the most likely solution given transcriptomic, proteomic or metabolomic data (reviewed in [11]). Among all available methods, the one from [12] seems promising for studies at the population/species level. It is based on the assumption that, at the genome scale, fluxes should covary with enzymatic protein abundances. Unlike the GIMME method [13], it takes into account quantitative variations instead of presence/absence. It does not make any assumptions about the shape of the flux-enzyme relationship like in IOMA [14]. Moreover, it does not requires heavy model curation nor absolute quantification of protein abundances like in Resource Balance Analysis [15, 16]. In [12], the only assumption is the expectation that there should be some correlation at the population level between fluxes and enzyme abundances. When considering quantitative proteomic data, which partly account for post-translational regulations, such assumption sounds biologically relevant. Irrespective of the method, comparisons rely on the probability distribution of the solution space, which is analytically intractable because of the stoichiometry constraints. Recently, [17] have proposed a Bayesian probabilistic method to characterize the solution space that is much faster than the classical Hit and Run algorithm [18] and allows for analyses at both the genome and population scale.

The HeterosYeast project investigated the molecular bases of heterosis in yeast at two different levels of integration: the proteomic level and the integrated trait level [19–21]. A diallel study of two yeast species involved in wine fermentation was carried out and the hybrid and parental strains were monitored during fermentation of grape juice at two temperatures. Integrated and proteomic traits were analyzed. In brief, the most important findings were: homeostasis of the interspecific hybrids observed at the trait level [21] and the predominance of interspecific heterosis at the proteomic level [20]. A closer analysis of genetic variance components confirmed that high-level phenotypes tended to exhibit higher additive genetic variance and lower interaction variance than proteomic traits [22]. However, previous studies on the HeterosYeast data set failed to find a clear link between variation at the trait level and variation at the proteomic level.

Given the important genomic resources in yeast [23], a number of curated genome-scale metabolic models are now available [24]. Among these, the DynamoYeast model [25] describes central carbon metabolism in yeast. It is small enough (70 reactions and 60 metabolites) to remain tractable, and has been tested against experimental data [26].

The availability of the HeterosYeast dataset, combined with a curated metabolic model of central carbon metabolism in yeast and a probabilistic approach to explore the solution space, encouraged us to integrate experimental proteomic data into the metabolic model in order to predict unobserved metabolic fluxes. We used these predicted fluxes to bridge the gap between proteomic data and integrated traits, and better understand the metabolic basis of life-history trait variation. This approach allowed us to show that the negative relationship between growth/fermentation traits and production traits is accounted for by a differential usage of the energy production pathway.

## Results

The HeterosYeast dataset provided valuable observations on the genetic diversity of yeast strains involved in the wine-making process at different levels of cellular organization, *i.e.* phenotypic traits related to life-history or fermentation [21], and quantitative proteomic data [20].

All traits were estimated or measured at 18˚C and 26˚C on a half-diallel design comprising 7 strains of *S. cerevisiae* and 4 strains of *S. uvarum*, with a total of 127 strain × temperature combinations.

In order to access an intermediate level of integration between protein abundances and traits, we used the DynamoYeast model, a curated Constraint-Based Model (CBM) of central carbon metabolism in yeast consisting of 70 reactions and 60 metabolites [25] (Fig 1). In the DynamoYeast model, the only entry is glucose, and the model does not take into account the complexity of metabolism like the recycling of amino-acids or extra-cellular TCA supplements. However, it was shown to accurately predict growth on complex medium like grape must [26]. Using gene-protein-reaction associations, enzymatic proteins and protein complexes were linked to the reactions of the CBM. Among the 70 enzymatic proteins and protein complexes, the abundances of 33 of them were retrieved from the dataset of 615 protein abundances quantified in the HeterosYeast project.

Thus, the metabolic fluxes that best matched the observed patterns of variation of enzymatic protein abundance were retained (see Material and methods). In brief, the strategy we proposed was: (i) to characterize the feasible solution space of the DynamoYeast model, *L*, through the posterior density distribution of the fluxes given by the Expectation Propagation algorithm (hereafter denoted EP, [17]); (ii) to select a unique solution through minimization of the objective function that measures the Euclidean distance between observed enzyme abundances and reaction rates (*Z*, Eq 11 in Material and methods).

Below, we first describe the method and its validation using simulated datasets. Then, we analyze the relationship between the different integration levels, using the HeterosYeast dataset and the predicted fluxes of central carbon metabolism.

## Sampling the feasible solution space with the expectation propagation algorithm

Sampling points from the feasible solution space *L* can be performed directly from the posterior truncated multivariate normal distribution of the fluxes. We compared the Hit and Run (HR) algorithm [27] with the EP posterior distribution of the fluxes to test the goodness of prediction of the EP algorithm [17] on the DynamoYeast posterior distribution.

We first characterized the feasible space of solutions *L* of fluxes from the DynamosYeast model from both methods (S1 File). The means of the posterior density distribution of fluxes obtained by the HR algorithm (burning length equal to $10^6$ and a jump of 0.5) for $10^6$ samples were well correlated with the means estimated with the EP algorithm ($r_{mean} = 0.98$), with moderate correlations between the variances ($r_{var} = 0.4$). The correlation between means and variances increased when considering $10^7$ samples ($r_{mean} = 1$, $r_{var} = 0.93$) (S1 and S2 Figs). We further observed that the EP algorithm well predicts the sampled space of solutions (S1 Fig) and the variance-covariance matrix (S3 Fig) between the DynamoYeast fluxes.

These results are similar to the ones obtained by [17]. Therefore, we decided to use the EP algorithm to sample the feasible solution space of the CBM.

## Protein abundances are good predictors of the initial set of metabolic fluxes

Computer simulations were performed to assess the goodness of prediction of the proposed method, as detailed in section Testing the prediction algorithm. The two main parameters that were tested were: *(i)* the number of sampled points $N_s$ in *L*; *(ii)* the number of observed proteins $N^{obs}$ to be included in the objective function *Z* (Eq 11 in Material and methods). To this end, a vector of flux values, $v^{initial}$, was first sampled from the feasible solution space of the

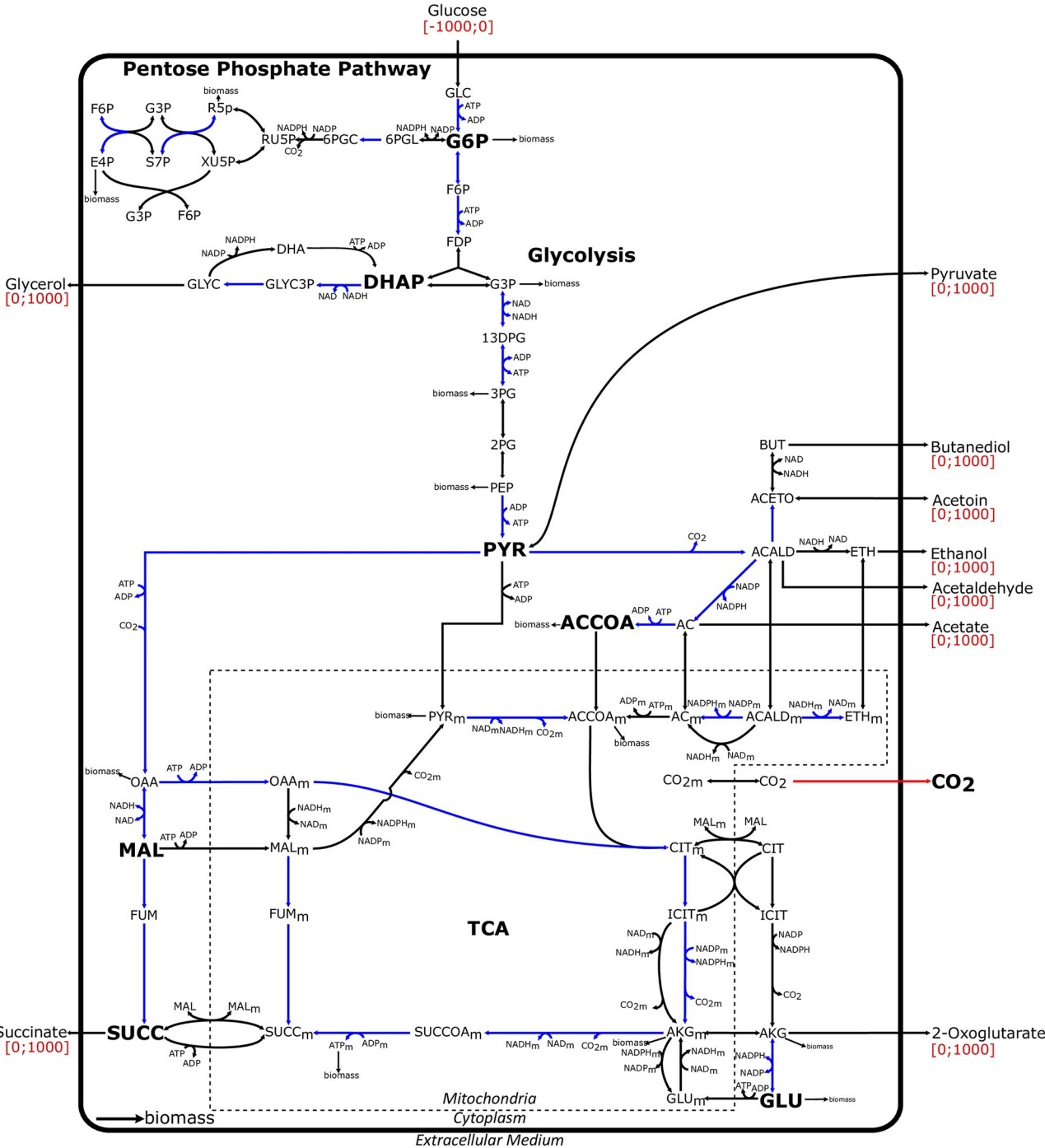

**Fig 1. Representation of the DynamoYeast model of central carbon metabolism in *S. cerevisiae*.** Metabolites are in black. Constraints on exchange fluxes are indicated in red between square brackets and correspond to fermentation, with glucose as the only input flux. The reaction color code is as follows: (*i*) in red, the experimentally measured $CO_2$ exchange flux; (*ii*) in blue, the reactions associated with enzymatic proteins quantified in this study; (*iii*) in black, the other reactions present in the DynamoYeast model. The list of the full metabolite names is in S1 Table.

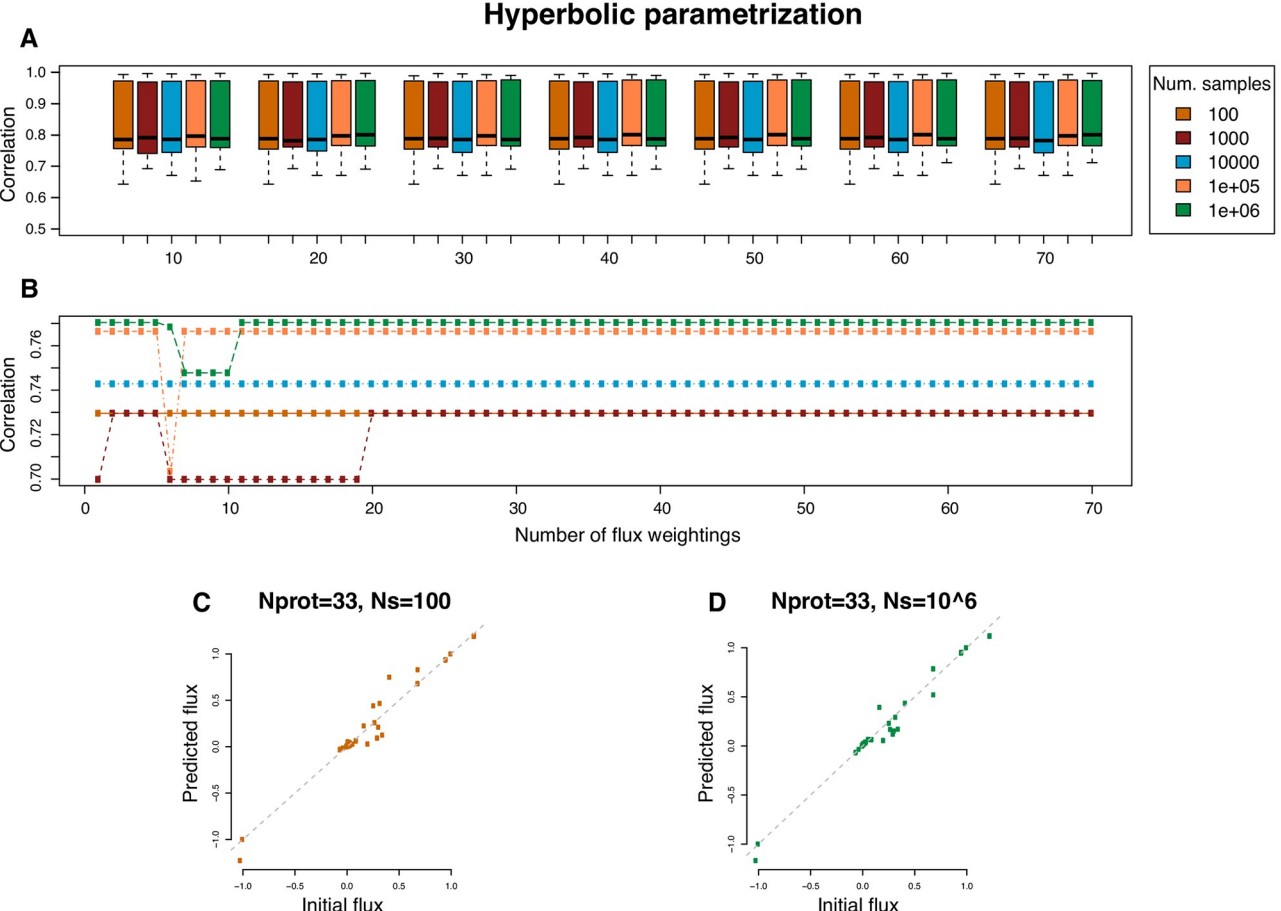

**Fig 2. Correlation between initial and predicted fluxes in simulated datasets using the DynamoYeast model.** Enzymatic protein abundances were expressed in terms of a hyperbolic function of the initial fluxes using Eq 12. Colors indicate the number of points $N_s$ that were sampled in solution space $L$. **A**. Boxplot representation as a function of the number of observed proteins $N^{obs}$, from 10 to 70, in increments of ten. Each box represents 1,000 simulations. **B**. Observed changes in the correlation during a single simulation run when the number of observed proteins was increased in increments of 1, from 1 to 70 (same color code as in **A**). The relationship between initial and predicted fluxes shown for one simulation with $N^{obs} = 33$ (matching the 33 enzymes of the HeterosYeast dataset) and $N_s = 100$ (**C**), and $N^{obs} = 33$ and $N_s = 10^6$ (**D**).

DynamoYeast model. We then computed protein abundances by assuming different functional relationships between fluxes and enzymatic proteins, and we used the proposed method to predict metabolic flux values, $v^{predicted}$. Different *a priori* functional relationships between fluxes and enzymatic proteins led to similar results. Simulations showed that minimization of $Z$ led to a high correlation between $v^{initial}$ and $v^{predicted}$ (Fig 2A). Correlations ranged from 0.65 to 0.99 (p-value $< 0.05$). By increasing the number of sampled points, $N_s$ in $L$, the mean correlation slightly increased and its variance decreased. The number of observed protein abundances $N^{obs}$ had a more complex influence on the accuracy of the predictions. When increasing $N^{obs}$, the correlation between $v^{initial}$ and $v^{predicted}$ either increased, decreased or stayed constant, as illustrated in Fig 2B. However, the order of magnitude of the variation was small, and the correlation tended to be more stable for high $N_s$ value (Fig 2B). In particular, the stability is reached when the number of observed proteins exceeds the size of the null space ($Ker(S) = 16$), *i.e.* the number of degrees of freedom of the DynamoYeast metabolic model.

When considering the actual set of enzyme abundances that were matched between the HeterosYeast proteomic data and the DynamoYeast CBM ($N^{obs} = 33$), we observed a high

correlation between $\mathbf{v}^{\text{initial}}$ and $\mathbf{v}^{\text{predicted}}$. This correlation increased when increasing the number of sampled points (see for example Fig 2C and 2D). We also checked that the 33 reactions were distributed over the modules of the metabolic reaction tree [28] (see Testing the CBM coverage with the observed proteomic dataset), as shown in S4 Fig. Altogether, we considered that our algorithm was efficient for predicting unobserved fluxes from enzyme abundances, given the correlation values obtained through simulations. More generally, simulations provide a method to check for the good coverage of the metabolic model with observations, without needing to predict elementary modes, which becomes computationally heavy as the size of the metabolic model increases.

## Predicting unobserved fluxes from the observed variation in protein abundances

The HeterosYeast proteomic data were used in the context of the DynamoYeast model of central carbon metabolism in yeast. In addition, for each strain × temperature combination, the observed $CO_2$ release rate was used as an additional constraint in the form of *a priori* knowledge to delimit the feasible solution space $L$. We sampled $N_s = 10^6$ points in $L$ to select a unique solution that minimizes the Euclidean distance between fluxes and enzyme abundances. As a result, we predicted 69 unobserved fluxes in the CBM for each of the 127 strain × temperature combinations. Statistical approaches were then used to investigate the variation components and the structure of the new dataset $D$ consisting of 615 protein abundances (**E**), 70 metabolic fluxes (**V**) and 28 fermentation and life-history traits (**T**):

$$D = (\mathbf{E}, \mathbf{V}, \mathbf{T})$$

## Patterns of variation depend on the integration level

The 127 observations in the new dataset $D$ had a specific structure. There were 7 parental strains (*S.c.*) and 21 intraspecific hybrids from *S. cerevisiae* (*S.c.* × *S.c.*), 4 parental strains (*S.u.*) and 6 intraspecific hybrids from *S. uvarum* (*S.u.* × *S.u.*), and 28 interspecific hybrids (*S.c.* × *S.u.*). All strains were observed during alcoholic fermentation of wine grape juice at two temperatures, 18˚C and 26˚C [21].

To better understand the patterns of variation at each integration level, Principal Component Analyses (PCA) were performed for each type of trait separately. Results are presented in Fig 3, where strains are identified by species, type of cross (intraspecific hybrid, interspecific hybrid or parental strain) and temperature. The first PCA components accounted for 20%, 23% and 27% of the total variation and the second for 14%, 18% and 19% of the total variation for protein abundances, metabolic fluxes and fermentation/life-history traits, respectively (Fig 3A, 3B and 3C). Different integration levels displayed different patterns of phenotypic diversity.

At the proteomic level (**E**), the first two PCA axes contributed to both differences between temperatures and between species and type of cross. Heterosis was observed for the three types of hybrids at both temperatures. First, *S.u.* × *S.u.* hybrids were clearly differentiated from their *S.u.* parents. Second, *S.c.* × *S.u.* interspecific hybrids were closer to their *S.c.* parents than their *S.u.* parents. Finally, *S.c.* × *S.c.* hybrids were close to their *S.c.* parents, but the range of variation between *S.c.* × *S.c.* hybrids was greater than between parental strains. Altogether, protein abundance in a hybrid strain cannot be predicted by the mean of its parental values.

At the trait level (**T**), we observed a high temperature effect, with axis 1 (27% of the variation) clearly separating the strains that were grown at 26˚C from those that were grown at 18˚C. At 26˚C, strains were characterized by high growth rates ($r$), high $CO_2$ fluxes ($J_{\max}$ and

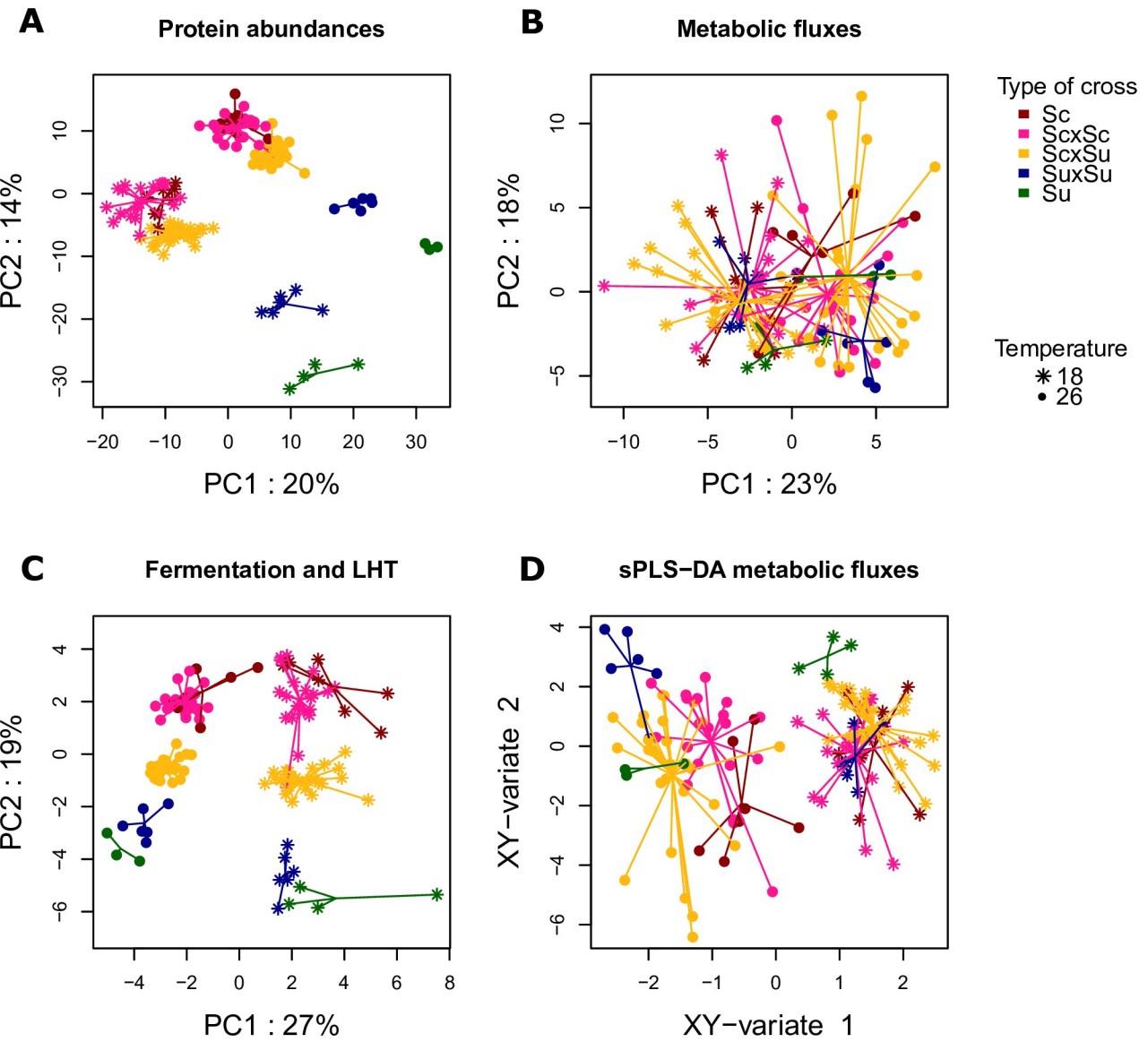

**Fig 3. Principal Component Analysis and sparce Partial Least Square-Discriminant Analysis.** PCA for protein abundances (A), metabolic fluxes (B) and fermentation/life-history traits (C). sPLS-DA for metabolic fluxes (D). Observations are represented on the first two PCA axes (sPLS-DA, respectively). Each dot corresponds to a strain × temperature combination. Different types of dots correspond to different temperatures, while the type of cross is color-coded. Segments join the data points to the centroid of the group (cross × temperature) to which they belong.

$V_{max}$), high *Hexanol* and *Decanoic acid*, a low carrying capacity (*K*) and short fermentation times (*AFtime*, *t-lag*, *t-75*, *t-45*, S5 Fig). At 18˚C, strains were characterized by low growth rates and low $CO_2$ fluxes, a high *K* and long fermentation times (S5 Fig). These two groups of traits mostly varied with temperature, although some differences between strains were observed within rather than between types of cross, especially at 18˚C. At 26˚C, *S.u.* strains perform slightly better than *S.c.* strains (higher growth rates, faster fermentation). The types of cross were clearly separated along PCA axis 2. Heterosis was observed at the trait level in intra-specific hybrids. However, interspecific hybrids seemed to be in-between the two parental strains. Traits that explain the differences between the observations along axis 2 were cell size (*Size-t-N_{max}*) and *Ethanol* at the end of fermentation (positively correlated with axis 2), aroma

production at the end of fermentation, as well as *Sugar.Ethanol.Yield* (negatively correlated) (S5 Fig). Note that these traits were not influenced by temperature. Hence, at the trait level, we observed differences between yeast species for traits related to aroma production that were not influenced by temperature. Most fermentation and life-history traits showed a strong temperature effect, large differences between strains within a type of cross, and weak heterosis.

At the flux level (**V**), temperature separated the observations on axis 1, but both axis 1 and axis 2 distinguished strains independently of their origin. Notice however that the variation range in hybrids was greater than in the parental strains, indicating differences between inbred and hybrid strains. Altogether, central carbon metabolic fluxes were influenced by temperature and showed strong differences between strains that were not related to the type of cross or the parental species. Sparse Partial Least Squares Discriminant Analysis (sPLS-DA) was performed on metabolic fluxes in order to select the main features that best characterize the species × temperature combinations (Fig 3D). As previously, the first axis distinguished strains grown at different temperatures. Six fluxes contributed to the first axis of the sPLS-DA: $CO_2$, ethanol, pyruvate decarboxylase, alcohol dehydrogenase, 6-phosphogluconolactonase and phosphogluconate dehydrogenase fluxes (S6 Fig). All were negatively correlated with axis 1 and were involved in fermentation. This shows that fermentation was more efficient at 26˚C. The second axis distinguished inbred strains from intraspecific hybrids with a genotype × temperature interaction: both *S.u.* × *S.u.* and *S.c.* × *S.c.* hybrids had higher coordinates than their parents at 26˚C, whereas *S.u.* × *S.u.* had lower coordinates than their parents at 18˚C, and *S.c.* × *S.c.* hybrids were confounded with their parental strains. Interspecific hybrids were characterized by a wide variation range at both temperatures. Fluxes that contributed to axis 2 were mainly mitochondrial fluxes. Mitochondrial acetyl-CoA formation, mitochondrial citrate synthase, mitochondrial aconitate hydratase, mitochondrial isocitrate dehydrogenase (NAD+) and mitocondrial transport fluxes of pyruvate, oxaloacetate and acetaldehyde were negatively correlated with the second axis, while mitochondrial transport of 2-oxodicarboylate, ethanol and $CO_2$ fluxes were positively correlated (S6 Fig).

In summary, we found at each integration level a strong temperature effect, large differences between strains, and evidence for heterosis, *i.e.* differences between hybrids and midparent values. However, patterns differed between the proteomic and the most integrated level. At the proteomic level, proteins involved in differences between strains were the same as the ones involved in differences between species and between temperatures. At the flux level, there were few differences between species. Differences between temperatures were associated with enzymatic reactions related to fermentation, while differences between strains were associated with enzymatic reactions that were either involved in fermentation, or in the part of the TCA cycle that takes place in mitochondria. At the trait level, differences between temperatures were associated with differences in growth and fermentation traits, which were relatively conserved within species but showed between-strain variation. Differences between species mostly concerned volatile compounds that are produced by secondary metabolism at the end of fermentation.

## Fermentation and life-history traits are associated with different metabolic pathways of carbon metabolism in yeast

Regularized Canonical Correlation Analysis (rCCA) was performed to investigate the correlation between metabolic fluxes and fermentation/life-history traits (Fig 4). Fermentation and life-history traits could be divided into two main groups showing contrasting profiles. The first group consisted of traits that clustered with the carrying capacity, *K*. They were negatively correlated with fluxes involved in the glycolysis, ethanol synthesis and pentose phosphate

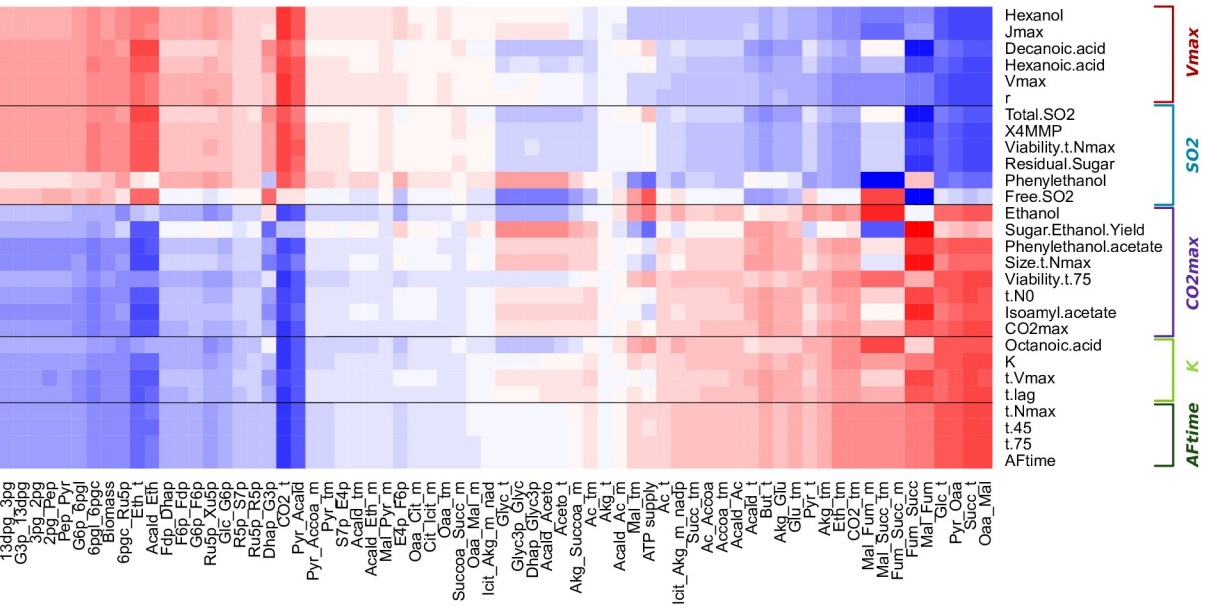

**Fig 4. Regularized Canonical Correlation Analysis of metabolic fluxes and fermentation/life-history traits.** Penalization parameters were tuned through a leave-one-out cross-validation method on a $1000 \times 1000$ grid between 0.0001 and 1 ($\lambda_1 = 0.8$, $\lambda_2 = 0.0001$). Canonical correlation values between metabolic fluxes and fermentation/life-history traits are represented as a gradient of colors from blue ($-1$) to red ($+1$). Metabolic fluxes (columns) and fermentation/life-history traits (rows) were clustered using the *hclust* method. Flux names are encoded as abbreviations of the substrate and the product of the reaction connected by "_". The five groups defined by fermentation and life-history traits are shown on the right.

pathways, and positively correlated with fluxes in the TCA reductive branch. By contrast, the second group consisted of traits that clustered with the intrinsic growth rate, *r*, and were positively correlated with fluxes involved in the glycolysis, ethanol synthesis and pentose phosphate pathways and negatively correlated with fluxes in the TCA reductive branch. Consistent with these results, the *biomass* pseudo-flux was positively correlated with *r* and negatively correlated with *K*.

When looking at the flux correlation structure described in Fig 4, we see the opposition between glycolytic fluxes and TCA fluxes. High growth rates and $CO_2$ fluxes ($J_{max}$, $V_{max}$) and correspondingly fast fermentation (short fermentation times) seem to be associated with a central carbon metabolism oriented towards fermentation, while high carrying capacity, low growth rate and slow fermentation seem to be associated with a central carbon metabolism oriented towards the production of downstream metabolites (succinate, pyruvate, acetate, acetaldehyde and butanediol).

The *K* group could be divided into three subgroups, based mainly on the correlation between traits and the glycerol synthesis and acetaldehyde fluxes: **AFtime**, **K** and **CO₂max** (subgroups designated by the name of the main trait in boldface). The **AFtime** subgroup showed a slightly negative correlation, the **K** subgroup a slightly positive correlation and the **CO₂max** subgroup a positive correlation. **AFtime** grouped most traits correlated with the duration of fermentation, *AFTime*, *t-45*, *t-75*, *t-N*$_{max}$; **K** grouped traits measuring the lag time and the beginning of fermentation (*t-lag*, *t-V*$_{max}$), the carrying capacity (*K*) and the concentration of *Octanoic acid* (a fatty acid) at the end of fermentation, while **CO₂max** grouped traits correlated with fermentation products (*total* $CO_2$, *Ethanol* and *Sugar.Ethanol.Yield*), two volatile esters, *Isoamyl acetate* and *Phenyl-2-ethanol acetate*, as well as cell size and cell viability measured close to the end of fermentation, and *t-N*$_0$.

Similarly, within the $r$ group we distinguished two clusters of traits: **SO$_2$** and **Vmax**. **SO$_2$** grouped basic enological parameters measured at the end of fermentation (*total and free SO$_2$*, *residual sugar*), cell viability measured once carrying capacity is reached (*Viability-t-N$_{max}$*), and two volatile compounds *Phenyl-2-ethanol* (alcohol) and *4-methyl-4-mercaptopentan-2-one* (*X4MMP*, thiol). **Vmax** grouped traits that correlated with $V_{max}$ and $r$, such as $J_{max}$ and the amount of *hexanol* (alcohol) and *hexanoic* and *decanoic acids* (fatty-acid) that were quantified at the end of fermentation (Table 1).

In summary, we were able to associate fermentation and life-history traits with metabolic fluxes based on their correlation patterns. In particular, we found that the negative correlation between $r$ and $K$ is explained by a different pathway usage of the central carbon metabolism. A high $r$ and a low $K$ are associated with glycolysis and fermentation, while a low $r$ and a high $K$ are associated with the TCA cycle and respiration.

**Table 1. Group name, trait description and abbreviation for each fermentation/life-history trait analyzed in this study.** Traits are grouped following the correlation structure obtained by rCCA (Fig 4).

| Group Name | Trait | Abbreviation |
|---|---|---|
| **AFtime** | Duration of fermentation | *AFtime* |
| | Time at which the carrying capacity is reached | *t-N$_{max}$* |
| | Fermentation time at which 45 gL$^{-1}$ of CO$_2$ was released, out of the fermentation lag-phase | *t-45* |
| | Fermentation time at which 75 gL$^{-1}$ of CO$_2$ was released, out of the fermentation lag-phase | *t-75* |
| **K** | Carrying capacity | *K* |
| | Fatty acid | *Octanoic acid* |
| | Time to reach the inflection point out of the fermentation lag-phase | *t-V$_{max}$* |
| | Fermentation lag-phase | *t-lag* |
| **CO2max** | Total amount of CO$_2$ released at the end of the fermentation | *CO$_{2max}$* |
| | Average cell size at *t-N$_{max}$* | *Size-t-N$_{max}$* |
| | Percentage of living cells at *t-75* | *Viability-t-75* |
| | Growth lag-phase | *t-N$_0$* |
| | Ratio between the amount of metabolized sugar and the amount of released ethanol | *Sugar.Ethanol.Yield* |
| | Basic enological parameter | *Ethanol* |
| | Ester | *Isoamyl acetate* |
| | Ester | *Phenylethanol. acetate* |
| **SO2** | Basic enological parameter | *Total SO$_2$* |
| | Basic enological parameter | *Free SO$_2$* |
| | Basic enological parameter | *Residual Sugar* |
| | Thiol | *X4MMP* |
| | Alcohol | *Phenylethanol* |
| | Percentage of living cells at *t-N$_{max}$* | *Viability-t-N$_{max}$* |
| **Vmax** | Maximum CO$_2$ release rate | *V$_{max}$* |
| | Maximum CO$_2$ production rate divided by the estimated cell concentration | *J$_{max}$* |
| | Growth rate | *r* |
| | Alcohol | *Hexanol* |
| | Fatty acid | *Decanoic acid* |
| | Fatty acid | *Hexanoic acid* |

## Metabolic bases of phenotypic trait variation in yeast

In order to confirm the association between integrated trait variation and the differential usage of central carbon metabolism, we identified proteins involved in trait patterning that were not included in the DynamoYeast model, as observed from the correlations between traits and fluxes (see section Statistical analysis). We performed a Linear Discriminant Analysis on the correlation matrix between the **T** traits and the **E** proteins using as discriminant features the five groups of fermentation and life-history traits described above. Linear Discriminant Analysis clearly separated the five trait categories along the first axis, which explains 99% of the total variation (Fig 5). **AFtime** and **K** traits were close to each other, and had positive coordinates on LDA1; **Vmax** had high negative coordinates, **SO₂** had a slightly negative mean and **CO2max** had a slightly positive mean on LDA1. Given the high discriminative power of LDA1, it is clear that proteins that were positively or negatively correlated with LDA1 participate in the differentiation of **AFtime** and **Vmax** trait groups.

Functional analysis of proteins that best correlate with the first axis of the LDA was performed on the group of proteins with a correlation of 0.85 in the positive and in the negative direction. Pearson's chi square test of enrichment showed that the group of proteins that were negatively correlated with the first axis was enriched in proteins linked to protein fate, cytoskeleton, detoxification, growth and death but also to the fermentation, glycolysis and phosphate pathways. The group of proteins that were positively correlated with LDA1 was enriched in proteins linked to energy conversion, nitrogen and sulfur pathways, metabolism,

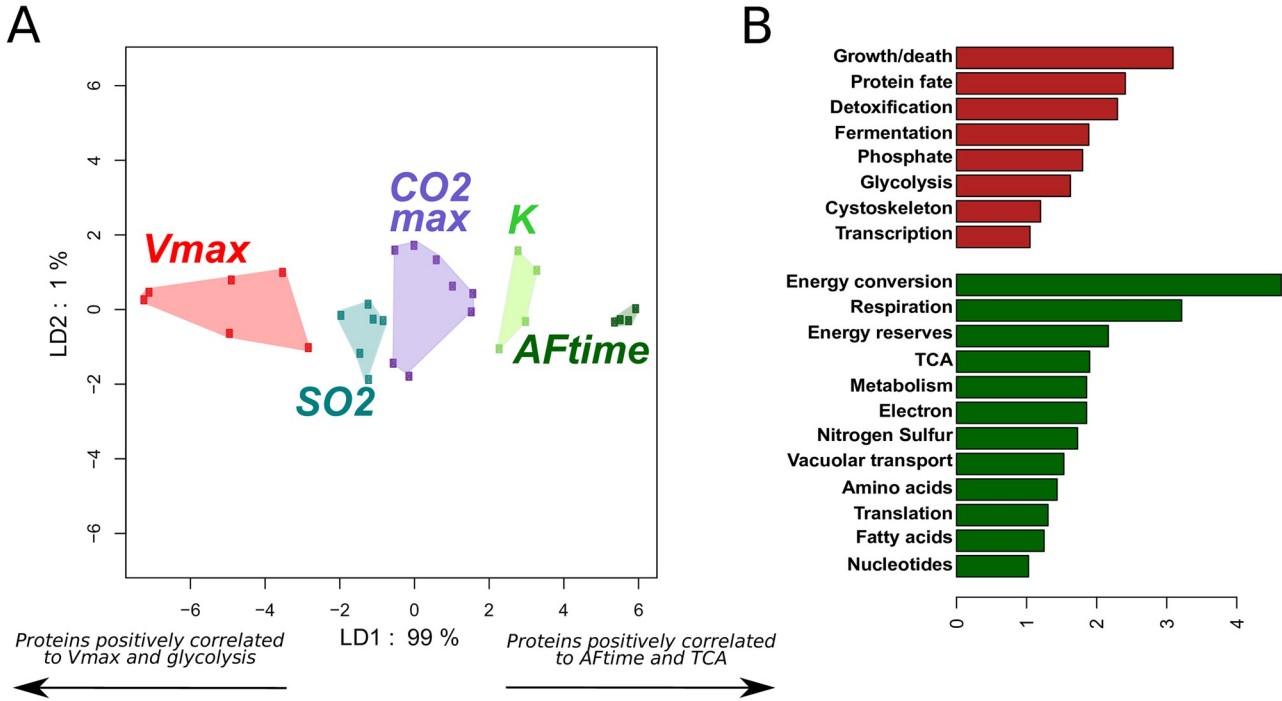

**Fig 5. Linear Discriminant Analysis of protein abundances. A**. Projection of the 28 fermentation/life-history traits onto the first two axes of a Linear Discriminant Analysis of protein abundances. Groups of traits were defined from their correlation with central carbon metabolism fluxes. Each dot corresponds to one fermentation or life-history trait. Colors correspond to groups of traits, which are named after one representative trait. The results confirm the structure of fermentation and life-history traits and reveal two groups of traits with antagonistic proteomic patterns: the **AFtime** group and the **Vmax** group. **B**. Functional enrichment categories of proteins that are positively (red) and negatively (green) correlated with the first LD1 axis. The groups of proteins were defined taking a positive (resp. negative) correlation threshold of 0.85 (resp. −0.85). The bars represent the proportion of proteins positively/negatively correlated to the LD axis belonging to a functional category divided by the proportion of proteins from the same category found in the MIPS database. Only the categories for which this value is higher than 1 are shown.

energy reserves, electron transport, TCA and respiration. This result was represented as a barplot in Fig 5.

In order to evaluate the interest of considering the flux level as an intermediate level between proteins and integrated traits, we run the same analysis by discarding the flux levels. We performed a rCCA to investigate the correlations between protein abundances and fermentation/life-history traits, and we identified seven groups of traits. We retrieved the **AFtime**, **CO2max** and **K** groups previously identified, but **Jmax** no longer belonged to the **Vmax** group, and trait related to the Phenylethanol pathway differentiate themselves from the **SO2** group. This time, the first two axes of the Linear Discriminant Analysis were needed to differentiate the seven groups of traits, with the first axis explaining only 79% of the total inter-groups variation (S7 Fig). While this pattern of variation clearly better illustrates the complexity of cell functioning and the importance of all metabolic reactions that fall outside the central carbon pathway, it fails to capture proteins directly connected to central carbon metabolism. On the contrary, the approach that consists in (i) defining groups of traits on the basis of their correlation with central carbon metabolic fluxes and (ii) seeking for additional proteins, outside central carbon metabolism, that differentiate the best between groups of traits, allowed us to track additional pathways that link central carbon metabolism to the variation of more integrated traits.

In conclusion, the association between trait variation and central carbon metabolism observed at the flux level is confirmed by the proteomic analysis. Proteins that covary with traits of the **Vmax** group and with glycolytic and fermentation fluxes are involved not only in glycolysis and fermentation, but also in protein synthesis and degradation (protein fate) and in cytoskeleton formation, which can be associated with cell division. Proteins that covary with traits of the **AFtime** group and with the TCA cycle and respiration fluxes are involved not only in the TCA cycle and respiration, but also in electron transport, energy conversion and nitrogen and sulfur metabolism.

## Discussion

We applied cutting-edge methods of data integration to an original yeast dataset. The HeterosYeast dataset comprised quantitative proteomic data as well as fermentation and life-history traits measured during wine fermentation on a range of strains from two yeast species. The objective was to integrate information at different levels of cellular organization (proteomic and metabolic fluxes) to better understand the metabolic bases of phenotypic variation in yeast, in particular life-history traits related to fitness. The key point of this study was to incorporate proteomic data in a constraint-based metabolic model to estimate the values of unobserved metabolic fluxes. Using a combination of multivariate analyses of heterogeneous high-dimensional datasets, we were able to show that the metabolic flux level retains information that is not directly interpretable at the proteomic or trait level. In particular, we showed that the negative correlation between traits associated with population growth rate and traits associated with maximum population size (carrying capacity) could be explained by a differential usage of central carbon metabolism, in this case fermentation *versus* TCA cycle.

Functional genome annotations, coupled with current knowledge in biochemistry, now allow cell metabolism to be described at the genome scale, using constraint-based metabolic models that take into account the stoichiometry of each reaction and incorporate thermodynamic constraints [29]. Without any *a priori* knowledge, the number of steady-state solutions for reaction rates is infinite; however, this number can be reduced by integrating observations. Three types of experimental data can be used: *(i)* exchange metabolic fluxes; *(ii)* metabolite input/output rates and *(iii)* protein abundances. External metabolic fluxes and metabolite

input/output rates can be used directly in constraint-based models to reduce the feasible solution space, $L$ (Eq 5 and inequality Eq 6) under the steady-state assumption.

Protein abundances, linked to the metabolic fluxes considered in the model through a gene-protein-reaction (GPR) association, carry information regarding network functioning and the state of the metabolic network at a given time and under specific conditions. Following [12], we used protein abundance profiles to find the set of metabolic fluxes that minimized the Euclidean distance between metabolic fluxes and enzyme abundances. Indeed, even though the relationship between flux and enzyme abundances is commonly non-linear, the extent to which a particular pathway is used is more or less associated with the abundance of its enzymes [30]. Despite the fact that the DynamoYeast metabolic model is an oversimplified model of central carbon metabolism with glucose as the only external carbon source, we show that protein abundance variations were sufficient to capture quantitative changes in the orientation of central carbon metabolism that occurred between strains and between growing temperatures in our dataset. Even though our flux predictions may not be very accurate, we are confident that we captured the main patterns of flux variation. Predicting unobserved fluxes from observed protein abundances overall adds information about the functioning of the actual metabolic network.

The method described here relies on a probabilistic approach. Following [17], we chose to characterize the feasible solution space $L$ by means of its posterior density distribution calculated with the Expectation Propagation (EP) algorithm. The computation time of the EP algorithm is much shorter than that of the well-known Hit and Run algorithm [18], allows to sample metabolic fluxes in $L$ and provides their associated posterior probability. In order to select a unique solution in $L$, we minimized $Z$, the Euclidean distance between observed protein abundances and the associated metabolic fluxes weighted by the inverse of the probability of observing such a set of fluxes, $p_v$ (Eq 11). This minimization process involved sampling in $L$, and selection was made after computing $Z$ over a high number of sampled points.

Computer simulations confirmed that our method had good prediction efficiency. In particular, we showed that prediction efficiency was not affected by the non-linearity of the flux-enzyme relationship. The most important parameter was the number of reactions $N^{obs}$ for which proteomic observations were available, compared to the CBM size, $n$. When $N^{obs}$ was too low, adding new information led to a decrease in prediction efficiency. A decrease in correlation between initial and predicted fluxes means that, when a new enzyme is added, the solution that minimized the total Euclidean distance can lead to flux predictions that are farther from their true value. This can occur whenever there is a weak correlation between the first $n-1$ fluxes, and the additional flux $v_n$. Therefore, it is important that protein abundance observations cover the main features of the architecture of the metabolic network. Here, we observed protein abundances for 33 out of the 70 reactions of the DynamoYeast model, which was sufficient to attain high prediction accuracy. The structure of the stoichiometry matrix allows us to define metabolic modules that correspond to main metabolic pathways [28]. Our simulations showed the importance of covering most metabolic modules with observations of protein abundance. Recent progress in gel-free/label-free quantitative proteomics now allow to quantify thousands of proteins and should ensure good coverage even for metabolic models at the genome scale [31].

In this study, we predicted the metabolic fluxes of central carbon metabolism in a population obtained from a half-diallel cross between two species of yeast, *S. cerevisiae* and *S. uvarum*. The genetic values of 615 protein abundances and 28 fermentation/life-history traits were estimated under fermentation conditions at two different temperatures, 18°C and 26°C, leading to a total of 127 observations from 66 different yeast strains [32]. As described above, we predicted metabolic fluxes for each strain × temperature combination by coupling the

DynamoYeast model, a highly curated Constraint-Based Model of central carbon metabolism [25] using the observed $CO_2$ release rate as *a priori* knowledge, and measurements of protein abundances associated with 33 out of the 70 reactions in the model.

The final dataset consisted of three matrices of $127 \times 615$ protein abundances, $127 \times 70$ central carbon fluxes, and $127 \times 28$ fermentation/life-history traits. As the total number of characters (713) greatly exceeded the number of samples (127), we used regularization techniques for the multivariate analyses [33]. In order to relate the variation patterns we observed at different integration levels, we used a top-down strategy from the most integrated to the least integrated level. First, we explored the correlation between traits and metabolic fluxes. Second, we identified the proteins which were not included in the metabolic model that best explained the correlation between traits and fluxes. We showed that the integration of the flux level allowed us to better understand the patterns of variation at the trait level.

In our dataset, we found a negative correlation between traits associated with growth and $CO_2$ fluxes, and traits associated with population size and fermentation length. These negative correlations reflected different life-history strategies, as has been observed previously in different yeast collections from either industrial [34] or natural sources [35, 36]. This broadly corresponds to the well-known *r-K* trade-off in ecology [37]. More recently, [38] suggested that such a trade-off could arise from eco-evolutionary feedback loops because competing strains also modify their environment through the production of different sets of metabolites. The HeterosYeast dataset shows that the choice of a strategy is plastic [21] and can be modified by the environment (here the fermentation temperature).

By adding information to the DynamoYeast model, we showed that such a trade-off can be explained by metabolic switches between fermentation associated with glycolysis, and downstream metabolite production, associated with the TCA cycle. This duality in the functioning of yeast central carbon metabolism was observed by [26], who matched the DynamoYeast model to experimentally measured exchange fluxes in different *S. cerevisiae* strains. The switch between the two modes of functioning (Fig 4) depends partly on the balance between two isoforms of alcohol dehydrogenase (ADH). Interestingly, [34] previously found that the trade-off between cell size and *K* is related to changes in the percentage of acetylation of ADH 1p, with high levels being associated with large cells and low *K*.

Because this paper describes a proof of concept, we deliberately chose to focus on central carbon metabolism and we used the DynamoYeast model because it contains a small number of reactions compared to available genome-scale models [24]. Therefore, we were not able to explain between-strain variation for traits related to secondary metabolism like aroma production, which merely discriminated between the two yeast species. Moreover, only a small subset of the proteomic data was coupled with the metabolic model. By searching for the proteins that best explained the trait patterns revealed at the flux level, we were able to identify proteins that were associated with the *r-K* trade-off at the trait level. Analysis of protein functional annotations confirmed the known link between the glycolysis and pentose-phosphate pathways and fermentation, and between extensive usage of TCA cycle and energy conversion.

Altogether, by coupling phenomic data with mathematical modeling of metabolism and cutting-edge statistical analyses (taking into account the high-dimensionality and heterogeneity of the measures), we were able to explain a well-known trade-off between two sets of yeast life-history traits by the differential usage of energy production pathways. Glycolysis and fermentation lead to fast growth and resource consumption. TCA and downstream metabolite production lead to slow growth and high population size. The duality between the two alternative uses of the central carbon metabolism is encoded into the architecture of the metabolic network.

## Material and methods

### Materials

**Materials: The HeterosYeast dataset.**    The genetic material for the experimental study consisted of 7 strains of *S. cerevisiae* and 4 strains of *S. uvarum* associated with various food processes (enology, brewing, cider fermentation and distilling) or isolated from the natural environment (oak exudates). The 11 parental lines were selfed and pairwise crossed, which resulted in a half-diallel design with a total of 66 strains: 11 inbred lines, 27 intraspecific hybrids (21 for *S. cerevisiae*, noted *S. c.* × *S. c.*, and 6 for *S. uvarum*, noted *S. u.* × *S. u.*) and 28 inter-specific (noted *S. c.* × *S. u*). The 66 strains were grown in triplicate in fermentors at two temperatures, 26˚C and 18˚, in a medium similar to enological conditions (Sauvignon blanc grape juice, [21]). From a total of 396 alcoholic fermentations (66 strains, 2 temperatures, 3 replicates), 31 failed due to the poor fermenting ability of certain strains. The design was set up as a block of two sets of 27 fermentations (26 plus a control without yeast to check for contamination), one carried out at 26˚C and the other at 18˚. The distribution of the strains in the block design was randomized to minimize residual variance of the estimators of the strain and temperature effects, as described in [34].

For each alcoholic fermentation, two types of phenotypic traits were measured or estimated from sophisticated data adjustment models: 35 fermentation/life-history traits and 615 protein abundances. Fermentation/life history traits were classified into four categories [21]:

- *Kinetics parameters*, computed from the $CO_2$ release curve modeled as a Weibull function fitted on $CO_2$ release quantification monitored by weight loss of bioreactors: the fermentation lag-phase, *t-lag* (h); the time to reach the inflection point out of the fermentation lag-phase, $t$-$V_{max}$ (h); the fermentation time at which 45 $gL^{-1}$ and 75 $gL^{-1}$ of $CO_2$ was released, out of the fermentation lag-phase, *t-45* (h) and *t-75* (h) respectively; the time between *t-lag* and the time at which the $CO_2$ emission rate became less than, or equal to, 0.05 $gL^{-1}h^{-1}$, *AFtime* (h); the maximum $CO_2$ release rate, $V_{max}$ ($gL^{-1} h^{-1}$); and the total amount of $CO_2$ released at the end of the fermentation, $CO_{2max}$ ($gL^{-1}$).

- *Life history traits*, estimated and computed from the cell concentration curves over time, modeled from population growth, cell size and viability quantified by flow cytometry analysis: the growth lag-phase, $t$-$N_0(h)$; the carrying capacity, $K$ (log[cells/mL]); the time at which the carrying capacity was reached, $t$-$N_{max}$ (h); the intrinsic growth rate, $r$ (log[cell division/mL/h]); the maximum value of the estimated $CO_2$ production rate divided by the estimated cell concentration, $J_{max}$ ($gh^{-1}10^{-8}cell^{-1}$); the average cell size at $t$-$N_{max}$, $Size$-$t$-$N_{max}(\mu m)$; the percentage of living cells at $t$-$N_{max}$, *Viability-t-$N_{max}$* (%); and the percentage of living cells at *t-75*, *Viability-t-75* (%).

- *Basic enological parameters*, quantified at the end of fermentation: *Residual Sugar* ($gL^{-1}$); *Ethanol* (%vol); the ratio between the amount of metabolized sugar and the amount of released ethanol, *Sugar.Ethanol.Yield* ($gL^{-1}$%$vol^{-1}$); *Acetic acid* ($gL^{-1}$ of $H_2SO_4$); *Total SO$_2$* ($mgL^{-1}$) and *Free SO$_2$* ($mgL^{-1}$).

- *Aromatic traits*, mainly volatile compounds measured at the end of alcoholic fermentation by GC-MS: two higher alcohols (*Phenyl-2-ethanol* and *Hexanol*, $mgL^{-1}$); seven esters (*Phenyl-2-ethanol acetate*, *Isoamyl acetate*, *Ethyl-propanoate*, *Ethyl-butanoate*, *Ethyl-hexanoate*, *Ethyl-octanoate* and *Ethyl-decanoate*, $mgL^{-1}$); three medium chain fatty acids (*Hexanoic acid*, *Octanoic acid* and *Decanoic acid*, $mgL^{-1}$); one thiol *4-methyl-4-mercaptopentan-2-one*, *X4MMP*($mgL^{-1}$) and the acetylation rate of higher alcohols, *Acetate ratio*.

For the proteomic analyses samples were harvested at 40% of $CO_2$ release, corresponding to the maximum rate of $CO_2$ release. Protein abundances were measured by LC-MS/MS techniques from both shared and proteotypic peptides relying on original Bayesian developments [39]. A total of 615 proteins were quantified in more than 122 strains × temperature combinations as explained in detail in [20].

**Genetic value of protein abundances and fermentation/life-history traits.** We considered the genetic value of protein abundances and fermentation/life-history traits, rather than their measured/computed value. In a previous study, [22] decomposed the phenotypic value of a trait at a given temperature, $P_T$, into its genetic, $G_T$, and residual, $\epsilon$, contributions:

$$P_T = G_T + \epsilon \tag{1}$$

The genetic value, $G_T$, was decomposed in terms of additive and interaction effects, taking into account the structure of the half-diallel design. By including two different species and the parental inbreds in the experimental design, we were able to distinguish between intra- and interspecific additive genetic effects ($A_w$ and $A_b$, respectively) and decompose the interaction effects into inbreeding ($B$) and intra and interspecific heterosis effects ($H_w$, $H_b$). Thus, the genetic value of a trait at a given temperature $T$ was modeled as follows:

$$G_T^{p_i} = \mu_T + 2A_{w_i,\ T} + \beta_{s(i),\ T} + B_{i,\ T} \tag{2}$$

$$G_T^{H_{ij}^w} = \mu_T + A_{w_i,\ T} + A_{w_j,\ T} + H_{w_{ij},\ T}, \tag{3}$$

$$G_T^{H_{ik}^b} = \mu_T + A_{b_i,\ T} + A_{b_k,\ T} + H_{b_{ik},\ T}. \tag{4}$$

for a parental strain $p_i$ (Eq 2), for an intraspecific hybrid $H_{ij}^w$ between parents $p_i$ and $p_j$ (Eq 3), and for an interspecific hybrid $H_{ik}^b$ between parents $p_i$ and $p_k$ (Eq 4). $\mu$ is the overall mean and $\beta_{s(i)}$ is the deviation from the fixed overall effect for the species:

$$s(i) \in \{S.\ cerevisiae, S.\ uvarum\}$$

We retrieved the genetic value for all proteomic data. For the fermentation traits, the model did not converge for most ethyl esters (*Ethyl-propanoate, Ethylbutanoate, Ethyl-hexanoate, Ethyl-octanoate* and *Ethyl-decanoate*), and for *Acetate Ratio* and *Acetic acid*. These traits were removed from the final analysis, which in the end included 28 traits.

**Protein functional annotation.** Cross-referencing MIPS micro-organism protein classification [40], KEGG pathway classification [41–43] and Saccharomyces Genome database [44] information, we attributed a single functional category to each protein.

The first two hierarchical levels of MIPS functional annotation were taken into account to assign proteins to 34 different categories. All secondary levels were used for the *01.metabolism, 02.energy* and *10.cell cycle and DNA processing* categories, resulting in 20 different functional categories. The *11.transcription* category was subdivided into the *transcription* sub-group (*11.06 and 11.02*) and into the *RNA processing* sub-group (*11.04*). Similarly, the *12.protein synthesis* category was split into the *ribosomal proteins* (*12.01*) and *translation* (*12.04, 12.07, 12.10*) sub-groups, and the *20.transport* category was split into the *vacuolar transport* (*20.09*) and *transport* (*20.01, 20.03*) sub-groups.

By contrast, the first hierarchical category was used for *14.protein fate, 30.signal transduction, 32.detoxification, 34.homeostasis, 40.cell growth and death, 42.cytoskeleton* In addition, *16. binding function* and *18.02.regulation* category were fused into *16.binding*, and *32.transposon*

*movement* was fused with *10.01.DNA processing*. Finally, *41.mating* and *43.budding* were included in the *10.03.cell cycle* category.

**DynamosYeast model.**    We used the DynamoYeast model, which is a previously developed Constraint-Based Model of central carbon metabolism in *S. cerevisiae* [25]. The main metabolic pathways included in this model are upper and lower glycolysis, the pentose phosphate pathway (PPP), glycerol synthesis, ethanol synthesis and the reductive and oxidative branches of the tricarboxylic acid (TCA) cycle. This model consists of 60 metabolites and 70 reactions, including one input flux, the uptake of glucose, and 10 output fluxes (Fig 1), taking place in the cytosol, mitochondria or in the extracellular medium.

The range of variation of the fluxes was fixed to allow for alcoholic fermentation. The following reactions were considered to be irreversible with $v^{inf} = 0$: *Oaa_Mal* (malate dehydrogenase), *Mal_Fum* (fumarase), *Fum_Succ* (fumarate reductase), and their respective mitochondrial counterparts *Oaa_Mal_m*, *Mal_Fum_m* and *Fum_Succ_m*, and *Oaa_Cit_m* (mitochondrial citrate synthase) (fluxes are denoted through the abbreviation of the substrate and the product connected by "_", followed by the enzyme name in parentheses). The fructose reaction was not included in the model, and *Glu_Akg_m* (mitochondrial glutamate dehydrogenase), as well as *Aceto_But* (butanediol dehydrogenase) were set to zero. In all, there were 16 reversible and 52 irreversible reactions.

Following the conventions implemented by many genome-scale metabolic models, many reactions of the DynamoYeast model of central carbon metabolism in *S. cerevisiae* are associated with genes and proteins via gene-protein-reaction (GPR) associations [45].

In general, there can be a many-to-many mapping of genes to reactions; for example, one reaction can be linked to proteins (*P1* and *P2*) or *P3*. The first Boolean AND relationship means that the reaction is catalyzed by a complex of two gene products. Since the maximum of the complex is given by the minimum of its components, the weighting of the complex is defined as: *P1* AND *P2* = min(*P1*, *P2*). The OR relationship allows for alternative catalysts of the reaction. Thus, total capacity is given by the sum of its components: (*P1* AND *P2*) OR *P3* = min(*P1*, *P2*) + *P3* [12]. Following these rules for each of the 11 yeast strains and the 55 hybrids at both temperatures, we estimated the protein abundances associated with the reactions of the DynamoYeast model, resulting in 33 reaction weightings out of 70.

Among the 70 reactions, only six were associated to protein complexes with an AND Boolean relationship, and four of them (in bold in the right column) matched two proteins of the HeterosYeast proteomic dataset (proteins in bold, Table 2).

We checked that pairwise correlations between proteins involved in the same complex were either positive or null, suggesting the absence of post-translational regulation that would change the overall enzyme concentration.

## Methods

**Constraint-based modeling of metabolic networks.**    Metabolic networks can be described in terms of the relationship between *M* metabolites, *m*, and *N* reactions, *v*, at a given time *t*:

$$(v, m)_t$$

Their topology can be expressed through the $M \times N$ stoichiometric matrix *S*, in which rows correspond to the stoichiometric coefficients of the corresponding metabolites in all reactions.

Under mass-balance assumptions and thermodynamic bounds of reaction rates, the dynamics of the network are governed by the linear system of constraints and inequalities:

$$Sv = \dot{m} \tag{5}$$

**Table 2. Reactions associated to protein complexes with an AND Boolean relationship.**

| Proteins | Reactions |
|---|---|
| (YOR136W AND YNL037C) | Icit_Akg_m_nad |
| (YIL125W AND **YDR148C** AND **YFL018C**) | **Akg_Succoa_m** |
| (**YGR240C** AND **YMR205C**) | **F6p_Fdp** |
| (YBR221C AND YER178W AND **YFL018C** AND YGR193C AND **YNL071W**) | **Pyr_Accoa_m** |
| (**YGR244C** AND **YOR142W**) | **Succoa_Succ_m** |
| (**YGL080W** AND (YGR243W OR YHR162W)) | Pyr_tm |

$$\boldsymbol{v}^{\mathrm{inf}} \leq \boldsymbol{v} \leq \boldsymbol{v}^{\mathrm{sup}} \tag{6}$$

where $\dot{\boldsymbol{m}} \in \mathbb{R}^M$ is the vector of the $M$ metabolite input/output rates, $\boldsymbol{v} \in \mathbb{R}^N$ is the set of $N$ reactions, and $\boldsymbol{v}^{\mathrm{inf}}$, $\boldsymbol{v}^{\mathrm{sup}}$ are the extremes of variation of the set of fluxes. Under a steady state assumption, $\dot{\boldsymbol{m}} = 0$ and the feasible solution space is expressed as:

$$L \equiv \left\{ \boldsymbol{v} \in \mathbb{R}^N | S\boldsymbol{v} = 0, \ \boldsymbol{v}^{\mathrm{inf}} \leq \boldsymbol{v} \leq \boldsymbol{v}^{\mathrm{sup}} \right\} \tag{7}$$

In general, $N$ is larger than $M$ and the solution space $L$ has infinite cardinality.

**Characterization of the feasible solution space.** We characterized the feasible solution space $L$ through the posterior probability of flux values obtained by the Expectation Propagation (EP) algorithm described in [17].

Instead of exploring $L$ by sampling, as classical methods do, [17] combined statistical physics and Bayesian approaches to infer the joint distribution of metabolic fluxes. To do so, given a set of metabolite input/output rates, $\dot{\boldsymbol{m}}$, they encoded the stoichiometric constraints within the likelihood posterior probability, defining a Boltzmann-like distribution with an energetic quadratic function

$$\mathcal{E}(\boldsymbol{v}) = \frac{1}{2} (S\boldsymbol{v} - \dot{\boldsymbol{m}})^{\top} (S\boldsymbol{v} - \dot{\boldsymbol{m}}) \tag{8}$$

while the inequality constrains were encoded in the prior probability of fluxes. Using Bayes theorem, this method provides a model for the posterior density of flux distribution.

Therefore, each point $\boldsymbol{v}$ in $L$ follows the truncated multivariate normal distribution

$$\forall \boldsymbol{v} \in L; \ \boldsymbol{v} \sim \mathcal{N}_T(\boldsymbol{\mu}, \boldsymbol{\Sigma} | \boldsymbol{v}^{inf}, \boldsymbol{v}^{\mathrm{sup}}, \dot{\boldsymbol{m}}) \tag{9}$$

where $\boldsymbol{\mu}$ is the vector of the mean posterior values of fluxes and $\boldsymbol{\Sigma}$ the posterior variance-covariance matrix of fluxes estimated with the EP algorithm.

For each set of metabolic fluxes $\boldsymbol{v}$, the posterior probability of observing $\boldsymbol{v}$ can be calculated at follows:

$$p_{\boldsymbol{v}} = P(\boldsymbol{v} | \boldsymbol{\mu}, \boldsymbol{\Sigma}, \boldsymbol{v}^{\mathrm{inf}}, \boldsymbol{v}^{\mathrm{sup}}, \dot{\boldsymbol{m}}) \tag{10}$$

Different values of extremes of variation can be used to model a particular process, for example for modeling reactions known to be irreversible in a specific context, *i. e.*

$$v_i^{\mathrm{inf}} = 0 \ or \ v_i^{\mathrm{sup}} = 0$$

or for introducing experimental data constraints, *i. e.*

$$v_i = v_i^{\mathrm{obs}} \pm \epsilon$$

for the *i*-th reaction.

Given that $\mu$ and $\Sigma$ depend on the imposed range $(v^{\mathrm{inf}}, v^{\mathrm{sup}})$ of internal and exchange fluxes, metabolic fluxes will take on particular values with probabilities that will depend on *a priori* knowledge and on the chosen metabolic process.

The algorithm implemented by [17] was translated into *R* code. Extraction of the stoichiometric matrix from the DynamosYeast model was performed with the *sybil* package in R [46].

**Prediction of metabolic fluxes from proteomic data.** In living systems, most metabolic reactions are catalyzed by enzymes, and quantitative proteomic data retain information about enzyme abundances. Therefore, the metabolism of a cell at a given time is characterized by the set of fluxes, metabolites and protein abundances

$$(\boldsymbol{v}, \boldsymbol{m}, \boldsymbol{E})_t$$

where $\boldsymbol{E} = (E_1, E_2, \ldots, E_N)$, and $E_i$ is the abundance of enzyme *i* associated with the reaction flux $\boldsymbol{v}_i$. Indeed, even though reaction rates are not directly proportional to enzyme abundances, a degree of covariation between protein abundance and flux reaction rate can be expected at the scale of the metabolic network. This can be used to infer intracellular metabolic fluxes with reasonable accuracy [12].

Among all possible solutions in *L*, we chose the one that minimized the objective function:

$$Z = \frac{1}{p_{\boldsymbol{v}}} \sum_{i=1}^{N} (E_i - |v_i|)^2 \tag{11}$$

*i.e.* the Euclidean distance between the observed protein abundances $\boldsymbol{E}_{obs}$ and the associated fluxes, weighted by $p_{\boldsymbol{v}}$, the posterior probability of observing the set of metabolic fluxes $\boldsymbol{v}$.

The properties of the truncated multivariate normal distribution ensure that the solution of the objective function is unique and no sophisticated algorithm is needed to find this solution. For each set of observations $\boldsymbol{E}_{\mathrm{obs}}$, we sampled $N_s$ points of the feasible solution space. Therefore, $\forall k \in \{1, 2, 3 \ldots N_s\}$, we obtained $\boldsymbol{v}^k \in L$ and $p_{\boldsymbol{v}^k}$. We calculated $Z^{(k)}$ and selected the set of flux values, $\boldsymbol{v}^{\mathrm{predicted}}$, for which $Z^{(k)}$ was the minimum.

In practice, it is never possible to associate each reaction of the metabolic network with a protein abundance. First, quantitative proteomics is not exhaustive. Second, reactions in a metabolic model are not always associated with an enzyme. Assuming a steady state condition and introducing information about protein abundances and measured external metabolic fluxes allows us to describe the system as:

$$(\mathbb{1}_{\mathrm{obs}} \boldsymbol{v} + \mathbb{1}_{\overline{\mathrm{obs}}} \, \boldsymbol{v}, \boldsymbol{m}_{\mathrm{const}}, \mathbb{1}_{\mathrm{obs}} \boldsymbol{E} + \mathbb{1}_{\overline{\mathrm{obs}}} \, \boldsymbol{E})_t \tag{18}$$

where $\mathbb{1}_{\mathrm{obs}}$ $(\mathbb{1}_{\overline{\mathrm{obs}}})$ is an indicator vector: its component-wise value is equal to 1 if the associated flux/protein component is observed (unobserved), otherwise it is equal to 0. Taking this into account, we reformulated the problem as follows:

- Observed fluxes were introduced as additional constraints with

$$v_i \sim \mathcal{N}(v_i^{\mathrm{obs}}, \sigma_{v_i}^2)$$

where $\sigma_{v_i}$ was set to a small value.

 

- The objective function was calculated only on the subset of observed enzyme abundances:

$$Z = \frac{1}{p_{\mathbf{v}}} \sum_{i=1}^{N^{\mathrm{obs}}} (E_i - |v_i|)^2$$

Predictions of metabolic fluxes were performed by coupling the DynamoYeast model to our experimental data (protein abundances and $CO_2$ reaction rates, the only measured flux in our study). We constrained the solution space $L$ by considering the maximum $CO_2$ release rate, measured at the same time point as the one used in the proteomics analyses [20]. For each strain × temperature combination a unique solution was obtained by minimizing the objective function, defined in Eq 11, from the set of observations.

**Testing the prediction algorithm.** The prediction algorithm is based on the assumption that fluxes and enzyme abundances covary. Indeed, any reaction rate can be expressed as a more or less complex function of enzyme abundances, kinetic constants and metabolite concentrations [47]:

$$v_i = k_{\mathrm{cat}_i} E_i f(\boldsymbol{\kappa}, \boldsymbol{m}, E)$$

where $k_{cat}$ is the catalytic constant, $\kappa$ is a set of other kinetic constants, $E$ is the set of enzymes abundances other than enzyme $i$. The $f$ function can be more or less complex depending on the mode of regulation.

To test the accuracy of the prediction of metabolic fluxes from protein abundance data, we used the feasible solution space of the DynamoYeast model and three different functions relating reaction rates to enzyme abundances. Specifically, we reversed the relationship, expressing protein abundance as a function of the reaction rate using a simplified formalism derived from the Metabolic Control Theory [48, 49]

$$v_{\mathrm{initial}} = \frac{1}{\frac{1}{A_i E_i} + \sum_{j \neq i} \frac{1}{A_J E_j}}$$

where the $A_j's$ are positive or negative constant terms. Given that enzyme concentrations cannot be negative, and taking $\forall j$, $A_j = \pm 1$, we obtain the hyperbolic relation:

$$E_i = \left| \frac{v_{\mathrm{initial}}}{1 - v_{\mathrm{initial}}} \right| \tag{12}$$

We also tested the predictions under the assumption that the relationship between protein abundances and flux reaction rates was linear:

$$E_i = k |v_{\mathrm{initial}}| \tag{13}$$

$k$ being an uniform random number $k \sim \mathcal{U}(0.1, 3)$

Finally, we considered the case where protein abundances and flux reaction rates are linked by a sigmoidal function [50], which we approximated with a Hill function:

$$E_i = \left| \frac{v_{\mathrm{initial}}^n}{1 - v_{\mathrm{initial}}^n} \right| \tag{14}$$

where $n$ is the Hill coefficient, sampled in the set $\Omega = \{2, 3, 4, 5\}$.

For each simulation, we sampled an initial set of fluxes $\boldsymbol{v}_{\mathrm{initial}} \in L$. We estimated the complete set of enzymatic protein abundances, $\boldsymbol{E}_{\mathrm{initial}}$ using Eqs 12, 13 or 14. Then, we minimized the objective function $Z$ to predict the set of fluxes $\boldsymbol{v}^{\mathrm{predicted}}$ that best fit enzyme abundances.

 

Prediction accuracy was measured as the correlation coefficient between $v^{\text{predicted}}$ and $v_{\text{initial}}$. Computer simulations were performed to test the influence of two main parameters: (*i*) the number of sampled points $N_s$; (*ii*) the number of quantified proteins, $N^{\text{obs}}$, included in the minimization process.

We assumed a steady state condition ($\dot{m} = 0$) and sampled $N_s$ points from the solution space of the multivariate posterior joint distribution of fluxes obtained using the EP algorithm [17]. We drew an additional point in solution space $L$, $v_{\text{initial}}$, and we calculated protein concentrations from the inverse problem. We retained the set of fluxes, $v^{\text{predicted}}$ for which $Z$ was minimum. The numbers $N^{\text{obs}}$ and $N_s$ were let to vary ($N_s \in \{10^2, 10^3, 10^4, 10^5, 10^6\}$ and $N^{\text{obs}} \in \{1, 2, 3\ldots\}$).

In terms of computational time, it would be expensive to consider all the possible combinations of observed enzymatic proteins associated with the metabolic model that can be included in Eq 11 (there are $N^{\text{obs}}(1 + (N^{\text{obs}} - 1) + (N^{\text{obs}} - 1)(N^{\text{obs}} - 2) + \cdots + (N^{\text{obs}} - 1)!)$ combinations). Therefore, for a given $N_s$, our strategy was to randomly choose one-by-one a protein to be included in the computation of the $Z$ function, and therefore for the prediction of metabolic fluxes $v^{\text{predicted}}$.

We randomly choose one reaction, $v_1$, from the complete set of reactions in the model, and we minimized

$$Z_1 = \frac{1}{p_v}(E_1 - |v_1|)^2 \tag{15}$$

to select one solution $v_1^{\text{predicted}}$ over the $N_s$ possible solutions of $L$. At the next iteration, we randomly chose an additional flux $v_2$ and its associated protein abundance $E_2$, and we minimized

$$Z_2 = \frac{1}{p_v}\sum_{i=1}^{2}(E_i - |v_i|)^2 \tag{16}$$

to predict $v_2^{\text{predicted}}$. This procedure was performed until the complete set of reactions was selected. In total, simulations were run a thousand times for different values of $N_s$ and $N^{\text{obs}}$.

**Testing the CBM coverage with the observed proteomic dataset.**   We performed a modular decomposition of the DynamoYeast model via the analysis of the null space (or kernel) of model stoichiometric matrix to check that the 33 reactions that matched with observations of protein abundances were distributed over the main pathways.

The stoichiometric matrix $S$ in the DynamoYeast model is of dimension $60 \times 70$ and of rank 54, meaning that 16 among the 70 reactions are strictly coupled. Thus, the dimension of the null space is $Ker(S) = 16$, roughly meaning that we need to know 16 independent flux combinations to predict changes in metabolic rates. Each reaction is associated with a 16 dimensional row vector in the null-space $70 \times 16$ matrix. We ran a hierarchical clustering using as a metric the symmetric matrix of the angles between the reaction vectors in the null-space [28] and we found, as previously shown, that the reactions cluster within metabolic pathways (S4 Fig). We found that the 33 reactions that matched with observations of protein abundances were distributed over the main pathways (S4 Fig).

In addition, we used our simulation method to check the correlations between initial and predicted fluxes by choosing as being observed the same 33 fluxes as in the experimental dataset.

## Statistical analysis

In order to study the main features characterizing fermentation and life-history traits in the HeterosYeast dataset, we analyzed the variation components of data from three different levels of cellular organization: protein abundances **E**, metabolic fluxes **V** and fermentation/life-history traits **T**:

$$D = (\mathbf{E}, \mathbf{V}, \mathbf{T})$$

The total number of observations was 127 strain × temperature combinations (66 strains, 2 temperatures, minus 5 missing data due to the poor fermenting abilities of some strains). The whole dataset consisted of 615 protein abundances, 70 metabolic fluxes and 28 fermentation and life-history traits. Two types of analyses were performed using different multivariate approaches: analyses at a single phenotypic level and analyses integrating the different levels.

We ran Principal Component Analyses (PCA) to identify the most important sources of variation within the datasets and the similarities/differences between the different phenotypic levels. We included prior knowledge regarding the yeast species in order to perform a supervised sparse Partial Least Squares Discriminant Analysis (sPLS-DA) to extract and combine discriminating features that best separate the different groups. The number of selected features was tuned using 3-fold cross-validation repeated 1,000 times.

In addition, the three levels of cellular organization were integrated in an unsupervised framework. First, we performed a regularized Canonical Correlation Analysis (rCCA) between fluxes **V** and integrated traits **T**, using the *mixomics* package in R [33, 51] to search for the key features that maximized the correlation between metabolic fluxes and fermentation traits. Second, since the correlation matrix between traits and fluxes was clearly structured, we computed the matrix of Euclidean distance between traits, based on their correlation with metabolic fluxes, and clustered traits using the *hclust* package in *R*. This procedure allowed us to define five groups of traits that showed similar correlation patterns with fluxes of the central carbon metabolism. Finally, we stored the linear correlation coefficients between proteins (P = 615) and traits (T = 28) in a ($T \times P$) matrix and ran a Linear Discriminant Analysis to search for proteins that best discriminate between groups of traits, considering traits as individuals. Pearson's chi-square test of enrichment was computed on protein functional category frequencies taking as prior probability the expected category frequency found in the MIPS database.

The same procedure was applied ignoring the flux level. rCCA was run between fluxes **V** and integrated traits **T**, and a hierarchical clustering was performed to group traits according to their correlations with protein abundances. Then a Linear Discriminant Analysis was performed to find the proteins that best discriminate between groups of traits.

## Supporting information

**S1 File. Sampling the solution space.** Comparison of the posterior density distribution obtained by Hit and Run (HR) sampling with the Expectation Propagation (EP) algorithm. (PDF)

**S1 Fig. Marginal probability densities of sixteen randomly chosen fluxes of carbon metabolism in yeast.** The histograms represent the HR result for $T \sim 10^7$ sampling points. The red line is the result of the EP estimate. (TIF)

**S2 Fig. Comparison of the results of HR versus EP.** The plots on the left are scatter plots of the means and on the right variances of the approximated marginals computed via EP

against the ones estimated via HR for an increasing number of explored configurations $T$, top $T \sim 10^6$, bottom $T \sim 10^7$.
(TIF)

**S3 Fig. Comparison of the results of HR versus EP.** The plot shows the relationship between 8 pairwise fluxes. Correlation ellipses computed by the EP algorithm are drawn in red. Dot points represent the mean value of fluxes computed with EP. For HR samples, $T \sim 5 * 10^6$.
(TIF)

**S4 Fig. Modular decomposition of the DynamoYeast model via null space analysis.** The null space of the stoichiometric matrix of the DynamoYeast model is spanned by the columns of the 70 reactions × 16 null-space matrix. Hierarchical clustering is applied using as a metric the angles between the 16 dimensional row vectors reactions. The reactions associated with enzymatic proteins quantified in this study are in blue, the other reactions present in the DynamoYeast model are in black.
(TIF)

**S5 Fig. Correlation between fermentation and life-history traits and the first two axes of the Principal Component Analysis.** The figure shows traits for which the correlation was > 0.5 or < −0.5 (p-value < 0.05). The first axis is negatively correlated with growth rate ($r$), $CO_2$ fluxes ($J_{\max}$ and $V_{\max}$), *Hexanol* and *Decanoic acid* and positively correlated with carrying capacity ($K$) and fermentation times (*AFtime, t-lag, t-75, t-45*). The second axis is positively correlated with cell size (*Size-t-$N_{\max}$*) and *Ethanol* at the end of fermentation, and negatively correlated with aroma production at the end of fermentation and *Sugar.Ethanol.Yield*.
(TIF)

**S6 Fig. Correlation between metabolic fluxes and the first two axes of the sparse Partial Least Square Discriminant Analysis.** The $CO_2$, pyruvate decarboxylase, ethanol, alcohol dehydrogenase, 6-phosphogluconolactonase and phosphogluconate dehydrogenase fluxes contributed to the first axis of the sPLS-DA, and were all negatively correlated with it. The second axis was negatively correlated with the mitochondrial acetyl-CoA formation, mitochondrial citrate synthase, mitochondrial aconitate hydratase, mitochondrial isocitrate dehydrogenase (NAD+) and mitochondrial transport fluxes of pyruvate, oxaloacetate and acetaldehyde fluxes, while positively correlated with the mitochondrial transport of 2-oxodicarboylate, ethanol and $CO_2$ fluxes.
(TIF)

**S7 Fig. Linear Discriminant Analysis of protein abundances using as discriminant features the groups identified on the basis of the correlations between integrated traits and protein abundances.** Projection of the 28 fermentation/life-history traits onto the first two axes of a Linear Discriminant Analysis of protein abundances. Each dot corresponds to one fermentation or life-history trait.
(TIF)

**S1 Table. Metabolites of the DynamoYeast model.** C and M stand for cellular and mitochondrial, respectively.
(XLSX)

## Author Contributions

**Conceptualization:** Marianyela Sabina Petrizzelli, Dominique de Vienne, Camille Noûs, Christine Dillmann.

**Data curation:** Thibault Nidelet.

**Formal analysis:** Marianyela Sabina Petrizzelli.

**Funding acquisition:** Marianyela Sabina Petrizzelli.

**Investigation:** Marianyela Sabina Petrizzelli, Christine Dillmann.

**Methodology:** Marianyela Sabina Petrizzelli, Christine Dillmann.

**Resources:** Thibault Nidelet.

**Supervision:** Dominique de Vienne, Christine Dillmann.

**Validation:** Dominique de Vienne, Christine Dillmann.

**Visualization:** Marianyela Sabina Petrizzelli.

**Writing – original draft:** Marianyela Sabina Petrizzelli.

**Writing – review & editing:** Marianyela Sabina Petrizzelli, Dominique de Vienne, Thibault Nidelet, Christine Dillmann.

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
