## [Decision Letter · Decision Letter 0]

30 Mar 2021

Dear Dr, Petrizzelli,

Thank you very much for submitting your manuscript "Data integration uncovers the metabolic bases of phenotypic variation in yeast" for consideration at PLOS Computational Biology.

As with all papers reviewed by the journal, your manuscript was reviewed by members of the editorial board and by several independent reviewers. In light of the reviews (below this email), we would like to invite the resubmission of a significantly-revised version that takes into account the reviewers' comments. Please pay particular attention to the points raised with respect to validation of your predictions, clarification of underlying assumptions and clear definition of the terminology used.

We cannot make any decision about publication until we have seen the revised manuscript and your response to the reviewers' comments. Your revised manuscript is also likely to be sent to reviewers for further evaluation.

Sincerely,

Christoph Kaleta

Associate Editor

PLOS Computational Biology

Jason Haugh

Deputy Editor

PLOS Computational Biology

Reviewer's Responses to Questions

**Comments to the Authors:**

Reviewer #1: The work by M. Petrizzelli et al. presents a new computational approach to estimate fluxes in the central metabolism of yeast based on proteomic data. Predictions are subsequently analyzed in the light of different yeast strains and hybrids from wine production that were previously described to show phenotypic variation in crucial life-history and fermentation traits. The authors conclude that differences in flux distributions display the underlying molecular basis underlying phenotypic variation.

##### Strengths

The presented study is solid work and has several notable strengths:

For a start, the authors present an interesting test case with proteomic data from a range of different yeast strains used for wine production. It is a strength of this study because it does not only include the best studied yeast strain S. cerevisiae but also a second, less studied, organism S. uvarum and hydrid strains.

Next, the work is based on a clearly described mathematical formulation of the flux prediction, which includes the analysis of the flux distribution solution space and the data-driven prediction the most likely distribution of fluxes based on enzymatic protein abundances. While the basic idea (predicting fluxes from proteomic data) is not per se novel, the authors describe a well-thought-out computational implementation and demonstrate its applicability to understand phenotypic variation in yeast, which will be of high interest to the scientific community studying yeast metabolism.

Moreover, data and scripts for the detailed quantitative data analysis are provided. Results are also discussed in detail by referring to relevant previous works.

I have a couple of points, which I invite the authors to consider in order to improve/clarify specific aspects of the manuscript:

##### Suggestions for manuscript improvement

Paragraphs at lines 41-58: It is not clear what the authors consider as high-throughput and what technique as low throughput. Also here, I am confident that many researchers would disagree that there are not current metabolomic approaches that can be considered high-throughput. E.g direct injection FTICR-MS provides you ten-thousands of masses and their intensities in less than a minute per sample. Thus when discussion metabolomics in the context here, the authors should refrain from using the term low/high-throughput but instead clearly describe the potential and shortcomming of metabolomis techniques to understand phenotypic variation on metabolic flux level.

Lines 55-56: "Technical developments in mass spectrometry have boosted metabolomics by enabling the characterization of the metabolome, i.e. the complete set of metabolites in a cell.". Please rephrase and clarify this sentence. First, why mentioning only mass spectrometry and not mass spectroscopy? Especially since NMR was mentioned just two sentences before in the same paragraph. Second, there is to date no technique that is able to characterize all metabolites in a cell; each technique can measure only a specific range of metabolites (e.g. with respect to a specific mass range, polarity, hydrophobicity, etc.).

Line 68-69. Why is this a specific populations genetics view? Isn't it more a cell physiology/evolutionary view?

Subsection 2.1 is very short. It begins by stating that the two algorithms (HT and EP) are compared; but it does not report any results from the comparison and only states that EP "gave a good approximation", without providing any quantitative results from the comparison. There's more detail in the appendix, but the reader would appreciate more details in the main text in order to understand the author's steps in this work.

A central notion in the manuscript is the distinction between "observable" versus "non observable" traits. Yet, the manuscript does not provide a clear definition for this distinction. For instance, are enzyme abundances "observable"; what about metabolite concentrations or reaction fluxes? Does non-observable mean that these traits are just difficult to measure?

The use of the term "secondary metabolites" is somehow different than in most publications. I am aware that in the scientific community "secondary metabolites" is loosely defined, but pyruvate, succinate and acetate are usually considered metabolites of the central metabolism and not secondary. Thus, I would use a different term than secondary metabolites (e.g. lines 217, 234, 362) to prevent misunderstandings. Perhaps, in the context of the present work, a term like "minor fermentation products" would be more fitting?

Fig. 5: Word clouds are not a scientifically sound way of presenting quantitative data, since visual differences might me misleading. Since the font size corresponds to the correlation of the respective fluxes in those groups with the LD1-axis, a better way to present this information would be a simple bar plot with the correlation value as bar height.

Fig. 1: To make this figure accessible to readers who are less accustomed to common metabolite identifiers, it would be nice if wherever possible to write the full metabolite names in the network and provide abbreviation information in the figure legends for all remaining metabolite IDs.

Figure 3D: Why is the x-axis scaled in a way that it shows ranges without data? If the scale is adjusted, difference between groups might be more obvious from the visualization.

Reviewer #2: The work by Petrizzelli er al. uses a constraint-based metabolic core model of S cerevidiae together with quantitative proteome data in order to predict metabolic flux distributions. These flux distributions parallel observations on the trait level and thus provide a rational and mechanistic interpretation.

In general, the work is interesting as it provides a data science approach to bridging disparate data sets. The presented work is sound, however, it’s main weak point is the lack of experimental validation. The authors aim to predict flux distributions in diverse yeast strains and confirm their validity indirectly by locking at phenotypic variation, but lack validation at the flux level (at least for some strains). Given the many simplifications applied I think it is necessary to provide an direct experimental validation at least for certain fluxes in selected strains to establish the feasibility of the suggested approach.

Major

* The authors simulate growth on minimal glucose limited medium and compare it to experimental data on chemical complex medium. Please justify that assumption. In particular, why do the authors not expect any impact from amino acid metabolism or extra cellular TCA supplements.

* The authors limit themselves to a core model of central carbon metabolism although for instance with yeast8 a highly curated metabolic model would be available too. It is even more surprising as the authors can therefore only use 33 protein abundance data of a much richer data set. This raises the concern that the observed correlation between the proteome and fluxome is a consequence of the very restricted degrees of freedom in the model. The authors should at least indicate the number of independent fluxes and the overlap with their proteome. In addition, the authors should enlarge their model and verify that the observed correlation remains similar.

* the authors strictly use the GPR mapping, in particular they use min(P1,P2) for an AND association. In their data, how often do the authors see that P1 is upregulated, while P2 is downregulated? This could be a hint at post-translational regulation at those points and should be at least mentioned. What if you exclude such data?

* L.95 “This approach …” I understand that this sentence refers to Fig 5 in particular. However, fig 5 is an LDA of Protein abundances. Thus, that conclusion could be drawn without use of the developed approach, couldn’t it?

* L.148 “algorithm was efficient for for” How was efficiency determined or measured? please define or reformulate

* L.294 “we were able to show that the metabolic flux level retains information” Please reformulate as you don#t know whether your predicted flux levels are correct.

* L.329 “Therefore, it is important that protein abundance observation cover the main features of the architecture …” This is a key point that I hinted above. However, the authors do not highlight how that can be achieved or what principle should govern that choice.

Minor

L.62, “The idea that a given set of environmental conditions will drive a cell to a steady state …” I think that is only true in the artificial setting of a chemostat but not true in any more realistic setting. Please reformulate.

L.65 “the number of metabolites is much higher than the number of reactions.” It’s the other way round in a (genome-scale) metabolic model

L..72 Please texpand your argument why [12] seems more promising a method than others. You say that fluxes should covary with enzyme abundance, essentially ignoring any post-translational regulation. Why should that be a realistic assumption?

Fig 1, CO2 secretion is measured, so no need to give [0:1000] as ranges

Fig 3, the two shades of blue are difficult to distinguish. please consider revising.

Out of interest, since your model is small, could you have done an elementar flux mode/vector analysis and characterised the totality of the solution space explicitly rather than doing sampling?

**Have all data underlying the figures and results presented in the manuscript been provided?**

Reviewer #1: Yes

Reviewer #2: Yes

PLOS authors have the option to publish the peer review history of their article (what does this mean?). If published, this will include your full peer review and any attached files.

Reviewer #1: **Yes: **Silvio Waschina

Reviewer #2: No
---

## [Decision Letter · Decision Letter 1]

7 Jun 2021

Dear Dr, Petrizzelli,

We are pleased to inform you that your manuscript 'Data integration uncovers the metabolic bases of phenotypic variation in yeast' has been provisionally accepted for publication in PLOS Computational Biology.

Best regards,

Christoph Kaleta

Associate Editor

PLOS Computational Biology

Jason Haugh

Deputy Editor

PLOS Computational Biology

Reviewer's Responses to Questions

**Comments to the Authors:**

Reviewer #1: The authors addressed all of my suggestions and concerns in this first round of revision. Especially, the presentation of the results was improved for instance in the text of subsection 2.1. and the figures 3 and 5. I am happy to recommend this work for publication.

Reviewer #2: my concerns have been addressed.

**Have the authors made all data and (if applicable) computational code underlying the findings in their manuscript fully available?**

Reviewer #1: Yes

Reviewer #2: Yes

PLOS authors have the option to publish the peer review history of their article (what does this mean?). If published, this will include your full peer review and any attached files.

Reviewer #1: No

Reviewer #2: No

---

## [Editor Report · Acceptance letter]

21 Jun 2021

PCOMPBIOL-D-21-00127R1 

Data integration uncovers the metabolic bases of phenotypic variation in yeast

Dear Dr Petrizzelli,

I am pleased to inform you that your manuscript has been formally accepted for publication in PLOS Computational Biology. Your manuscript is now with our production department and you will be notified of the publication date in due course.

With kind regards,

Zsofi Zombor
